# Effects of Clonidine as an Adjuvant to Lidocaine with Epinephrine in Ultrasound Guided Axillary Brachial Plexus Block: A Randomised Controlled Trial

**DOI:** 10.3390/jcm10184181

**Published:** 2021-09-16

**Authors:** Anil Ranganath, Tomas Hitka, Gabriella Iohom

**Affiliations:** 1Department of Anaesthesia and Intensive Care Medicine, Connolly Hospital, D15 X40D Dublin, Ireland; 2Department of Anaesthesia and Intensive Care Medicine, Cork University Hospital, T12 DC4A Cork, Ireland; hitka@yahoo.com (T.H.); giohom@ucc.ie (G.I.)

**Keywords:** local anaesthetic, adjuvant, clonidine, axillary brachial plexus block

## Abstract

This study evaluated the effects of adding adjuvant clonidine to lidocaine with epinephrine on the characteristics of ultrasound-guided axillary brachial plexus block (ABPB) for upper extremity surgery. Twenty-four patients were randomised to receive an ultrasound guided ABPB with 20 mL of lidocaine 2% with 1:200,000 epinephrine plus 2 mL of either normal saline 0.9% (Group 1) or a mixture of clonidine 1 µg/kg and normal saline 0.9% (Group 2). The outcome measures that were recorded were the overall onset time and the duration of sensory and motor block. The median (IQR) overall onset time of sensory and motor block was significantly shorter in Group 2 vs. Group 1 (5 (5–7.5) min vs. 10 (8.8–12.5) min; *p* < 0.001) and (5 (2.5–7.5) min vs. 7.5 (6.3–7.5) min; *p* = 0.001), respectively. The median (IQR) overall duration of sensory and motor block was significantly longer in Group 2 vs. Group 1 (225 (200–231) min vs. 168 (148–190) min; *p* < 0.001) and (225 (208–231) min vs. 168(148–186) min; *p* < 0.001), respectively. In ultrasound-guided ABPB, the addition of clonidine to lidocaine with epinephrine resulted in shorter onset time and prolonged duration of sensory and motor block.

## 1. Introduction

Adjuncts to local anaesthetics for peripheral nerve blocks have been widely used to enhance the quality and duration of both anaesthesia and postoperative analgesia [1,2,3,4,5,6]. The two most commonly used α_2_-adrenergic agonists to enhance the characteristics of nerve block are clonidine and dexmedetomidine [7,8]. Previous studies have yielded conflicting results when clonidine, an α_2_-adrenergic agonist, was combined with local anaesthetics for the purpose of a peripheral nerve blockade. While several studies on brachial plexus block have demonstrated shorter block onset time and longer anaesthesia duration [9,10,11,12,13,14,15], other studies have found contrasting results [16,17].

However, all of the aforementioned studies comparing block onset time and duration following the perineural injection of clonidine and local anaesthetic admixture used both conventional volumes and techniques for locating nerves. A question arises as to the effect of clonidine as an adjuvant to local anaesthetics in the context of ultrasound guidance and relatively lower volumes as per current practice.

In this prospective study, we set out to evaluate the effects of adding both clonidine and epinephrine to lidocaine 2% on the block onset time and the duration of ultrasound-guided axillary brachial plexus block. We hypothesised that using a 20 mL mixture of lidocaine 2% plus epinephrine 1:200,000 combined with clonidine 1 µg/kg would shorten the onset time of sensory block when compared to lidocaine 2% plus epinephrine 1:200,000 in ultrasound-guided axillary brachial plexus block (USgABPB) for upper limb trauma surgery.

## 2. Materials and Methods

This prospective, randomised, single-centre study was approved by The Clinical Research Ethics Committee of Cork Teaching Hospitals (ECM 4(aa) 04/03/14, Chairperson Professor Michael G Molloy), registered at https://clinicaltrials.gov (accessed on 5 July 2017) (NCT03207022), and conducted at Cork University Hospital. Written informed consent was obtained from all eligible participants. Twenty-four patients aged 18 years of age or older, ASA grade I-III;, undergoing unilateral upper limb surgeries of the forearm and hand, were recruited to the study. The exclusion criteria were contraindication to regional anaesthesia, hypersensitivity to amide local anaesthetics or clonidine, intolerance or contraindication to non-steroidal anti-inflammatory drugs, BMI > 35, pregnancy, cardiac conduction abnormalities, history of hepatic or renal failure, and neurological or neuromuscular disease.

On arrival to the anaesthesia induction room, an intravenous cannula was placed in the forearm contralateral to the surgical site, and standard monitors were applied. Using a computer-generated sequence of random numbers and a sealed envelope technique, the patients were randomised to one of two groups to receive USgABPB with 20 mL of lidocaine 2% plus 1:200,000 epinephrine combined with either 2 mL of normal saline 0.9% (Group 1) or 2 mL of an admixture of clonidine 1 µg/kg and normal saline 0.9% (Group 2). The patients were not made aware of group allocation. All of the blocks were performed by an experienced anaesthesiologist skilled in USgABPB and who was blinded to the group allocation, which ensured uniformity in the deposition of the local anaesthetic solution. The patients were positioned supine with the operative arm abducted and externally rotated with the elbow flexed at 90°. Under aseptic precautions, the axillary brachial plexus block was performed under ultrasound guidance alone using a SonoSite Titan unit (SonoSite^®^, Bothwell, WA, USA) with a 38 mm linear array 5–10 MHz transducer (L38). All four terminal branches, the median, ulnar, radial, and musculocutaneous nerves were identified in the axillary region. A 24-gauge, 50 mm, insulated short bevel needle (Stimuplex^®^ B. Braun, Melsungen, Germany) was advanced in-plane until the needle tip was placed adjacent to the nerve before the local anaesthetic was injected to produce a circumferential spread around each target nerve. The total volume of the local anaesthetic solution corresponding to the study group was equally distributed among the four nerves.

Sensory and motor blockade was evaluated after completion of the injection by an independent observer not aware of group allocation. Assessment of sensory and motor block onset in the innervation area of each nerve (median, ulnar, radial, and musculocutaneous nerve) were conducted every 2.5 min until surgical anaesthesia was achieved or after 30 min had elapsed. Sensory function was scored as being present or absent, and motor function was graded using the modified Bromage Scale (Table 1) [18]. Surgical anaesthesia was defined as a motor score ≤ 2 with an absent sensation to cold (tested with ethyl chloride BP, Cryogesic^®^, Dr Georg Friedrich Henning, Chemische Fabrik Walldorf GmbH, Walldorf, Germany). Each nerve distribution area was individually assessed, and the sensory and motor onset time was measured separately from the conclusion of the block (removal of block needle, T_0_) to the attainment of an absent sensation to cold and a motor score ≤ 2, respectively. The overall sensory and motor block onset time was taken from T_0_ to the attainment of surgical anaesthesia in all innervation territories. If surgical anaesthesia had not been achieved at 30 min in one or more of the four nerve distribution areas, a rescue block or general anaesthesia was planned. Any patients with discomfort or pain during surgery requiring supplementation of infiltration by surgeons or requiring opiate analgesia were considered block failure, and data from these patients were analysed separately. In case of patient anxiety or upon request, sedation with midazolam to a maximum of 3 mg was provided. All patients received paracetamol 1 g and diclofenac sodium 75 mg iv intraoperatively. Heart rate, blood pressure, peripheral oxygen saturation, and sedation score on a five-point scale (0 = wide awake, 1 = drowsy, 2 = dozing intermittently, 3 = mostly asleep, and 4= only aroused by tactile stimulation) were recorded every 5 min intraoperatively and every 30 min postoperatively until the resolution of the block. Hypotension and bradycardia, defined as a 20% fall in blood pressure and heart rate, respectively, in relation to pre-block baseline values and any episodes of Spo_2_ equal to or less than 90% associated with sedation requiring oxygen supplementation by venturi mask were noted.

Postoperatively, the duration of the sensory and motor block of each nerve was evaluated every 15 min by an independent observer. The time interval was taken from the completion of the block procedure to the return of sensation to cold and motor power (score ≥ 3), respectively. Postoperative analgesia consisted of regular paracetamol 1 g po every 6 h, diclofenac 75 mg po every 12 h, and rescue analgesia was offered in the form of oxycodone 10 mg orally every 4–6 as required.

The primary outcome was the overall onset of sensory block, which was defined as the time elapsed from the conclusion of the block (T_0_) until the attainment of sensory block in all four nerve distribution areas. Similarly, the overall onset of motor block was defined as the time interval from T_0_ to the attainment of motor power (score ≤ 2) in all four nerve distribution areas. Secondary outcome measures included the overall onset of the motor block, the onset of the sensory and motor block for individual nerves, the overall duration of the sensory and motor block, the time to the first request for postoperative opioid analgesia, and the incidence of adverse effects perioperatively.

### Sample Size and Statistical Analysis

The sample size was calculated based on the onset of sensory block as the primary outcome parameter. Kaabachi et al. [19] found a mean onset of sensory block of 9 (SD ± 3) min following an axillary brachial block performed with 30 mL lidocaine 1.5%. The minimum sample size required to have an 80% probability of detecting a 40% decrease in onset time (level of significance 0.05) was nine patients per group. We recruited 12 patients per group to account for potential dropouts.

Statistical analysis was performed using SPSS version 24 (IBM, Armonk, NY, USA). The Shapiro–Wilk test was used for normality testing. Continuous, normally distributed data are presented as mean (SD), and non-normally distributed data are presented as median (interquartile range (IQR)). Comparisons between the groups were analysed using the unpaired Student’s t test for normally distributed data, and nonparametric data were analysed with the Mann–Whitney U test. Categorical variables were compared between groups using Pearson’s or Fischer’s exact test. All tests were 2-tailed, and *p* < 0.05 was considered statistically significant.

## 3. Results

Twenty-four patients (12 in each group) were recruited to the study from April 2014 to January 2015. All patients completed the study (Figure 1), and none of the patients required rescue block, conversion to general anaesthesia, supplementation by surgeons, or intraoperative opioid analgesia. The patient demographic characteristics were similar between the groups (Table 2). The overall onset of sensory and motor block was significantly shorter in Group 2 compared to Group 1 (Table 3). This was also noted in the individual nerve block onset times, with the exception of musculocutaneous motor block. The overall duration of the sensory and motor block was significantly longer in Group 2 compared to Group 1 (Table 4). In addition, the overall duration of both the sensory and motor blocks of individual nerves were also longer in Group 2. Figure 2 depicts the primary outcome measure, which is the overall onset of sensory block.

Haemodynamic parameters (Figure 3, Figure 4 and Figure 5), peripheral oxygen saturation, and sedation score showed no significant differences between the groups. In seven patients, the highest sedation score was one, four of whom were in Group 1, and three of whom were in Group 2, during the intraoperative period, which corresponded to receiving midazolam for anxiety. There were no systemic adverse events noted in either group. Twelve patients, eight in Group 1 and four in Group 2, requested additional opiate analgesia postoperatively. The median (IQR) time to the first request for postoperative opioid analgesia was longer in Group 2 at 318 (303–469) min compared to 209 (166–268) min in Group 1 (*p* = 0.04).

## 4. Discussion

This study demonstrated that adding clonidine to lidocaine with epinephrine for ultrasound-guided axillary brachial plexus block resulted in shorter overall onset time, prolonged duration of both sensory and motor block, and prolonged duration of postoperative analgesia. Our results are similar to previous findings, whereby clonidine added to local anaesthetics in conventional axillary brachial plexus block shortened block onset time [9,10,11,12] and prolonged the duration of anaesthesia and analgesia following the conclusion of the blocks [9,10,11,12,13,14]. In contrast to earlier studies, we performed all of the blocks solely under ultrasound guidance, ensuring uniformity in the deposition of local anaesthetic around all four terminal nerves, thus minimising the differences associated with the technique.

The use of lidocaine, a local anaesthetic with a moderate duration of action, allowed the detection of any effects attributable to clonidine in the postoperative period. The dose of 1 µg/kg of clonidine was chosen based on previous reports [10,14]. Doses of up to 150 µg have been used with minimal side effects [20,21]. In a dose finding study, Bernard et al. demonstrated that the addition of clonidine to lidocaine 1% resulted in a more pronounced sensory blockade as well as in a dose-dependent prolongation of analgesia [10]. They concluded that 30 to 90 µg clonidine improved the nerve block quality while limiting the side effects. Similarly, Iohom et al. [11] demonstrated that the addition of clonidine to mepivacaine in an axillary brachial plexus block led to a decrease in sensory block onset time and an increase in the duration of anaesthesia and postoperative analgesia. More recently Hrishi et al. [22] found that clonidine added to a mixture of bupivacaine and lidocaine shortened the onset time and prolonged the duration of ultrasound-guided supraclavicular block. Of note, the heterogeneity of nerve stimulation (single or multiple) or the dual ultrasound plus nerve stimulation-based block techniques used may have influenced the spread of the local anaesthetic within the brachial plexus sheath and the subsequent block characteristics.

The addition of clonidine to local anaesthetics for brachial plexus block has been found to be efficacious when compared to the systemic administration of similar doses in prolonging the block duration and postoperative analgesia, suggesting a local mechanism of action of clonidine [23,24]. However, the precise mechanism of how clonidine exerts its action remains speculative. Several theories have been postulated, including one that states that that clonidine with selective α_2_ adrenoreceptor agonist and weak α_1_ agonist activity causes vasoconstriction by postsynaptic adrenoreceptor activation [25,26,27], thus prolonging block duration by reducing the vascular absorption of local anaesthetic mixtures. Other studies have not supported those results [16,28] and have hypothesised that clonidine may have a direct effect on nerve fiber conduction, mainly in A alpha and C fibres [29,30,31]. Despite this observed effect, clonidine alone has been shown to be incapable of producing analgesia when injected into the axillary brachial plexus sheath [32].

In the present study, all of the patients received lidocaine with epinephrine, which produces marked vasoconstriction when used alone. Whether this effect is further enhanced or prolonged by the addition of clonidine remains unknown. It is likely that the prolongation of local anaesthetic block occurs due to a combination of pharmacokinetic and pharmacodynamic effects on local anaesthetic actions [33].

Our study is limited by the moderate sample size. As it was not designed to elucidate the mechanism of action of clonidine, the study did not include a systemic clonidine group. This study adds to the body of evidence supporting the use of clonidine as an adjuvant to local anaesthetics in improving the efficacy and characteristics of peripheral nerve blocks.

## 5. Conclusions

In conclusion, an admixture of clonidine 1 µg/kg to lidocaine with epinephrine for ultrasound-guided axillary brachial plexus block resulted in the shorter onset time and the longer duration of both sensory and motor block as well as longer time until the first request for postoperative analgesia.

## Figures and Tables

**Figure 1 jcm-10-04181-f001:**
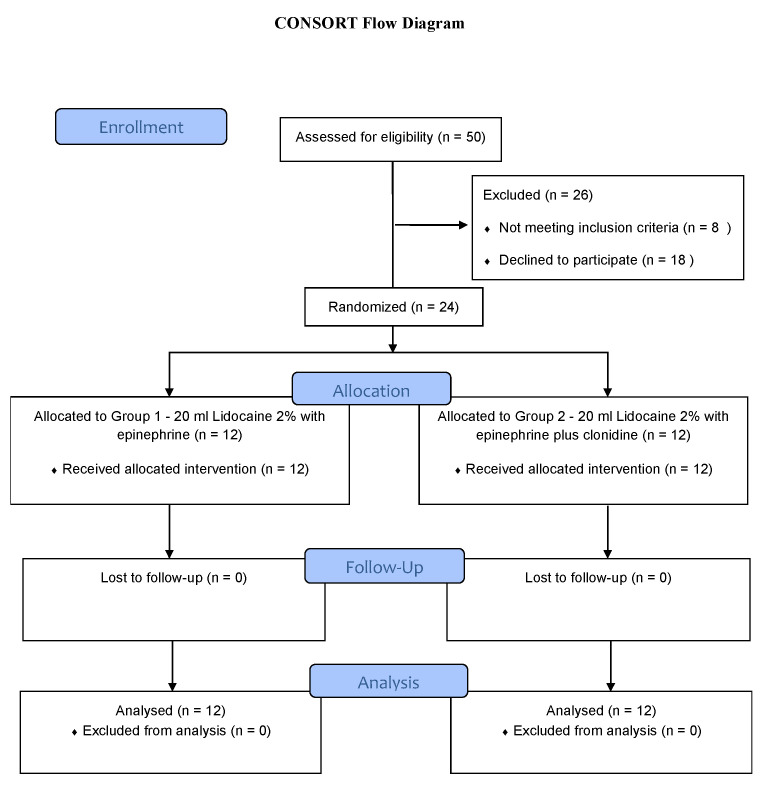
CONSORT patient flow diagram. *n* = number.

**Figure 2 jcm-10-04181-f002:**
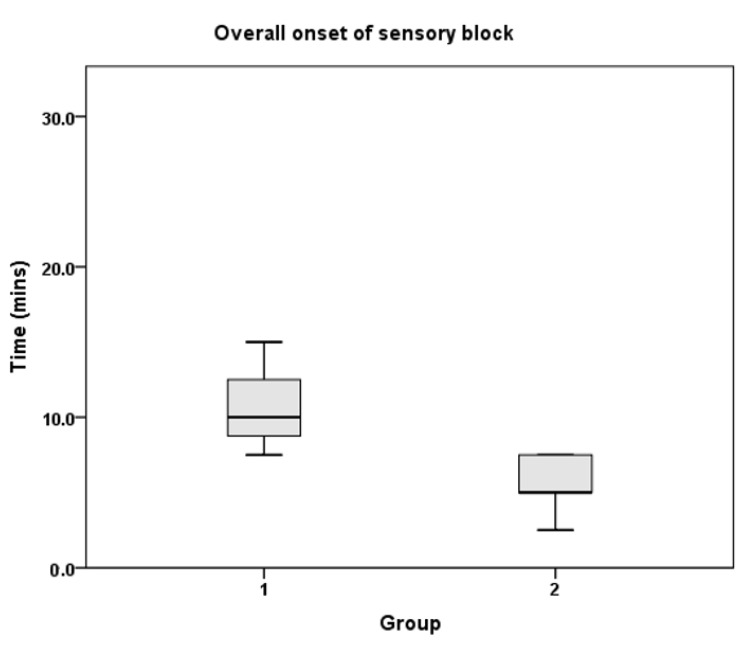
Overall onset of sensory block.

**Figure 3 jcm-10-04181-f003:**
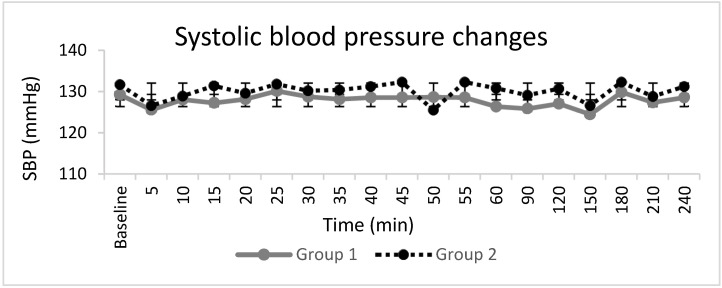
Systolic blood pressure changes (SBP). The line graphs represent the mean and standard deviation in Group 1 and Group 2.

**Figure 4 jcm-10-04181-f004:**
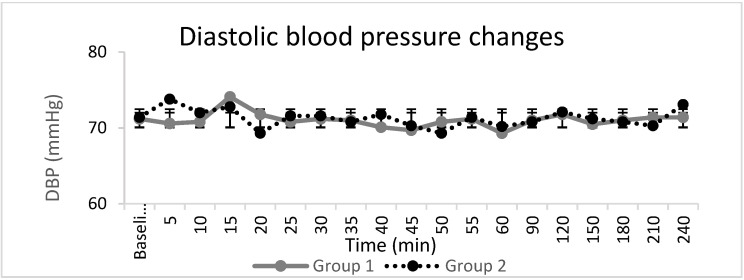
Diastolic blood pressure changes (DBP). The line graphs represent the mean and standard deviation in Group 1 and Group 2.

**Figure 5 jcm-10-04181-f005:**
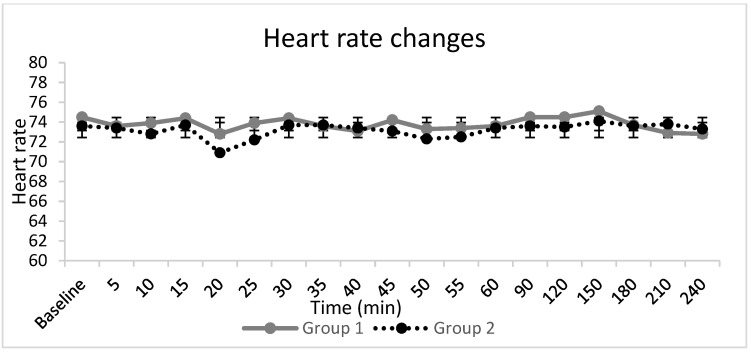
Heart reate changes. The line graphs represent the mean and standard deviation in group 1 and 2.

**Table 1 jcm-10-04181-t001:** Motor and sensory testing.

	Motor Test	Sensory Test
Median	Flexion of radial 3 fingers	Thenar eminence
Radial	Extension of wrist	Dorsum of hand
Ulnar	Abduction of fingers	Hypothenar eminence
Musculocutaneous	Elbow flexion	Over base first metacarpal
**Modified Bromage Scale** Score and definition 4 Full power in relevant muscle. 3 Reduced power but ability to move muscle against resistance 2 Ability to move relevant muscle group against gravity but not against resistance 1 Flicker of movement in relevant muscle group 0 No movement in relevant muscle group		

**Table 2 jcm-10-04181-t002:** Patient characteristics.

	Group 1 *n* = 12	Group 2 *n* = 12	*p* Value
Age, y	41.4 ± 10.6	42.3 ± 8.6	0.82
Sex, M/F,	7/5	8/4	0.67
BMI, Kg/m^2^	25.8 ± 1.8	25.7 ± 2.0	0.98
ASA grade (I/II/III), *n*	9/3/0	7/5/0	0.39
Duration of surgery, min	63.9 ± 15.9	61.8 ± 15.4	0.74
Site of surgery (Forearm, wrist, hand), *n*	0/10/2	1/6/5	

Continuous variables are presented as means ± SD and categorical variables as counts.

**Table 3 jcm-10-04181-t003:** Sensory and motor block onset time (min).

	Group 1 *n* = 12	Group 2 *n* = 12	*p* Value
Overall sensory block	10 (8.8–12.5)	5 (5–7.5)	<0.001
Overall motor block	7.5 (6.3–7.5)	5 (2.5–7.5)	0.001
Radial nerve			
Sensory block	10 (7.5–11.3)	5 (3.8–7.5)	0.002
Motor block	5 (5–7.5)	3.8 (2.5–5]	0.014
Ulnar nerve			
Sensory block	7.5 (7.5–10)	5 (5–6.3)	0.005
Motor block	5 (5–7.5)	3.75 (2.5–5)	0.007
Median nerve			
Sensory block	7.5 (5–11.25)	5 (5–7.5)	0.001
Motor block	5 (5–7.5)	3.8 (2.5–5)	0.005
Musculocutaneous nerve			
Sensory block	7.5 (7.5–8.8)	5 (5–6.8)	0.003
Motor block	5 (2.5–6.3)	2.5 (2.5–5)	0.18

Data are presented in minutes; values are median (interquartile range, Q1–Q3).

**Table 4 jcm-10-04181-t004:** Sensory and motor block duration (min).

	Group 1 *n* = 12	Group 2 *n* = 12	*p* Value
Overall sensory block	168 (148–190)	225 (200–231)	<0.001
Overall motor block	168 (148–186)	225 (208–231)	<0.001
Radial nerve			
Sensory block	175 (148–192)	225 (200–231)	0.001
Motor block	180 (151–187)	225 (208–231)	<0.001
Ulnar nerve			
Sensory block	182 (156–192)	227 (200–241)	0.002
Motor block	178 (148–190)	226 (211–233)	<0.001
Median nerve			
Sensory block	180 (156–190)	230 (200–241)	0.001
Motor block	176 (156–193)	225 (215–238)	<0.001
Musculocutaneous nerve			
Sensory block	180 (156–195)	233 (215–241)	0.001
Motor block	168 (156–195)	225 (215–231)	<0.001

Data are presented in minutes; values are median (interquartile range, Q1–Q3).

## Data Availability

Hard copies of the data details are available in the Department of Anaesthesia and Intensive Care Medicine, Cork University Hospital, Ireland.

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
