# Peer review of "Effects of Clonidine as an Adjuvant to Lidocaine with Epinephrine in Ultrasound Guided Axillary Brachial Plexus Block: A Randomised Controlled Trial"

_jcm, 2021, doi:10.3390/jcm10184181_

Round 1

Reviewer 1 Report

In the present manuscript, the authors have investigated the effects of clonidine as adjuvant to local anesthetic for peripheral nerve block. To clarify the effect of clonidine, the authors have planned prospective, randomized, single center study. The authors have shown that clonidine has shorter onset effect, and prolonged duration of block effects.

I have some points which the authors should refer to

Major Point

In the manuscript, the authors have used clonidine for alpha2-adrenergic agonist. Recently, dexmedetomidine is useful for clinical situation. There are papers comparing dexmedetomidine and clonidine for adjuvant to local anesthetics. The authors should mention using dexmedetomidine for adjuvant.

Minor point

In the manuscript, the authors have written "all the blocks were performed by an experienced anesthesiologist skilled in USgABPB".

Did the anesthesiologist know the Group1 or Group 2? This information should be included in the text.

Author Response

Thank you for your valuable time reviewing our submission and providing suggestions.

We have now revised the above manuscript in light of the reviewers’ comments.

Specifically, we have made the following changes highlighted in the text of the manuscript.

Response to Reviewer 1 Comments

Comments and Suggestions for Authors

In the present manuscript, the authors have investigated the effects of clonidine as adjuvant to local anaesthetic for peripheral nerve block. To clarify the effect of clonidine, the authors have planned prospective, randomized, single centre study. The authors have shown that clonidine has shorter onset effect, and prolonged duration of block effects.

I have some points which the authors should refer to

Point 1 (Major point): In the manuscript, the authors have used clonidine for alpha2-adrenergic agonist. Recently, dexmedetomidine is useful for clinical situation. There are papers comparing dexmedetomidine and clonidine for adjuvant to local anaesthetics. The authors should mention using dexmedetomidine for adjuvant.

Response 1: In agreement with the reviewer suggestion.

We are familiar with the numerous studies comparing the effectiveness of time tested α-2 adrenoreceptor agonist clonidine to newer α-2 agonist dexmedetomidine which has been cited in the review article.

El-Boghdadly K, Brull R, Sehmbi H, Abdallah FW. Perineural Dexmedetomidine Is More Effective Than Clonidine When Added to Local Anesthetic for Supraclavicular Brachial Plexus Block: A Systematic Review and Meta-analysis. Anesthesia and analgesia. 2017;124(6):2008-20.

We tried to address an important unmet need by carrying out original and valid piece of research with clonidine. As aforementioned studies in our article compared block characteristics following perineural injection of clonidine and local anaesthetic admixture, using both conventional volumes and techniques for locating nerves. The question arises as to the effect of clonidine as adjuvant to local anaesthetics in the context of ultrasound guidance and relatively lower volumes as per current practice.

However, we acknowledge the importance and clinical effectiveness of the newer α-2 adrenoreceptor agonist dexmedetomidine as an adjuvant to local anaesthetics to be mentioned, but we were limited by the scope and  the design of the study.

In response, additional of sentence in the introduction paragraph along with the references.

  1. El-Boghdadly K, Brull R, Sehmbi H, Abdallah FW. Perineural Dexmedetomidine Is More Effective Than Clonidine When Added to Local Anesthetic for Supraclavicular Brachial Plexus Block: A Systematic Review and Meta-analysis. Anesthesia and analgesia. 2017;124(6):2008-20.

  1. Vorobeichik L, Brull R, Abdallah FW. Evidence basis for using perineural dexmedetomidine to enhance the quality of brachial plexus nerve blocks: a systematic review and meta-analysis of randomized controlled trials. British journal of anaesthesia. 2017;118(2):167-81.

Point 2 (Minor point): In the manuscript, the authors have written "all the blocks were performed by an experienced anaesthesiologist skilled in USgABPB".

Did the anaesthesiologist know the Group1 or Group 2? This information should be included in the text.

Response 2: Thank you for notifying it.

As mentioned  in the paragraphs under the heading ‘ Materials and Methods’ It’s a computer-generated sequence of random numbers and a sealed envelope technique. Patients were not made aware of group allocation.  Independent observer unaware of group allocation , evaluated the sensory and motor blockade after completion of the injection. However, all blocks were  performed by experienced anaesthesiologist skilled in USgABPB, blinded to group allocation, ensuring uniformity in the deposition of local anaesthetic solution.

We have added the sentence in the paragraph.

We thank reviewer for their comments and greatly improving the manuscript.

Yours sincerely,

Anil Ranganath

Reviewer 2 Report

This is a single center, prospective, randomized, controlled, study, with the objective to study the "Effects of clonidine as an adjuvant to lidocaine with epinephrine in ultrasound guided axillary brachial plexus block" on 24 patients, allocated to received a brachial plexus block with lidocaine 1% with epinephrine and normal saline (Group 1) or the admixture of lidocaine with epinephrine 1% and clonidine 1 mcg/kg (Group 2), in both groups the block was ultrasound-guided., which allowed a more effective administration of the local anesthesia admixture. The study's  primary outcome was overall onset of sensory and motor block, which was defined as the time elapsed from conclusion of block until attainment of sensory block in all four nerve distribution areas. overall onset of motor block, onset of sensory and motor block of individual nerves, overall duration of sensory and motor block,
time to first request of postoperative opioid analgesia and incidence of adverse effects perioperatively.

Major Comments:

1) Most references induced in the manuscript were published more than 15-20 years ago, although several studies regarding the same topic and similar methodology have been published in recent years, and the vas majority showed similar results favoring the use of clonidine to shorten the onset of the brachial plexus block and increasing the duration of the block.

2) Authors did not addressed in the discussion the disparity of the results in regards the requirements of opioid analgesia between groups. 

Minor Comments:

  • Table 4 requires a better layout

Author Response

Thank you for your valuable time reviewing our submission and providing suggestions.

We have now revised the above manuscript in light of the reviewers’ comments.

Specifically, we have made the following changes highlighted in the text of the manuscript.

Response to Reviewer 2 Comments

Comments and Suggestions for Authors

This is a single centre, prospective, randomized, controlled, study, with the objective to study the "Effects of clonidine as an adjuvant to lidocaine with epinephrine in ultrasound guided axillary brachial plexus block" on 24 patients, allocated to receive a brachial plexus block with lidocaine 1% with epinephrine and normal saline (Group 1) or the admixture of lidocaine with epinephrine 1% and clonidine 1 mcg/kg (Group 2), in both groups the block was ultrasound-guided., which allowed a more effective administration of the local anaesthesia admixture. The study's  primary outcome was overall onset of sensory and motor block, which was defined as the time elapsed from conclusion of block until attainment of sensory block in all four nerve distribution areas. overall onset of motor block, onset of sensory and motor block of individual nerves, overall duration of sensory and motor block,
time to first request of postoperative opioid analgesia and incidence of adverse effects perioperatively.

I have some points which the authors should refer to

Point 1 (Major point): Most references induced in the manuscript were published more than 15-20 years ago, although several studies regarding the same topic and similar methodology have been published in recent years, and the vast majority showed similar results favouring the use of clonidine to shorten the onset of the brachial plexus block and increasing the duration of the block.

Response 1: In agreement with the reviewer suggestion.

We have added the following references in the article.  

  1. Patacsil JA, McAuliffe MS, Feyh LS, Sigmon LL. Local Anesthetic Adjuvants Providing the Longest Duration of Analgesia for Single- Injection Peripheral Nerve Blocks in Orthopedic Surgery: A Literature Review. AANA journal. 2016;84(2):95-103.

  1. Prabhakar A, Lambert T, Kaye RJ, Gaignard SM, Ragusa J, Wheat S, et al. Adjuvants in clinical regional anesthesia practice: A comprehensive review. Best practice & research Clinical anaesthesiology. 2019;33(4):415-23.

  1. Chawda PM, Sharma G. A clinical study comparing epinephrine 200μg or clonidine 90μg as adjuvants to local anaesthetic agent in brachial plexus block via supraclavicular approach. Journal of anaesthesiology, clinical pharmacology. 2010;26(4):523-7.
  2. Hrishi AP, Rao G, Lionel KR. Efficacy of Clonidine as an Additive on the Duration of Action of Brachial Plexus Block Performed Under Ultrasound and Nerve Locator Guidance: A Prospective Randomized Study. Anesthesia, essays and researches. 2019;13(1):105-10.
  1. Point 2 (Major point): Authors did not address in the discussion the disparity of the results in regards the requirements of opioid analgesia between groups. 

    Response 2: Thank you for constructive comments. Time to first requirement of postoperative opioid analgesia was significantly prolonged in the clonidine group. This was reflective of prolonged duration of anaesthesia due to addition of clonidine, hence the duration of analgesia.

    In agreement with the reviewer, we have added the sentence in the discussion paragraph. 

    Point 3 (Minor point) : Table 4 requires a better layout

    Response 3: In agreement with the reviewer. Tables 3 and 4 layout have been modified.

    We thank reviewer for their comments and greatly improving the manuscript.

    Yours sincerely,

    Anil Ranganath